# Analysing multimodal data that have been collected using photovoice as a research method

Roisin Mooney ⬤ ,[1] Kamaldeep Bhui,[1,2,3] Co-Pact Project Team

RM and KB are joint first authors.

<sup>1</sup>CHiMES Collaborative, Department of Psychiatry, University of Oxford, Oxford, UK
<sup>2</sup>Nuffield Department of Primary Care Health Sciences, Wadham College, University of Oxford, Oxford, UK
<sup>3</sup>Global Policy Institute, Queen Mary University London, London, UK

**Correspondence to**
Dr Roisin Mooney;
roisin.mooney@psych.ox.ac.uk

## ABSTRACT

**Background** Creative arts practice can enhance the depth and quality of mental health research by capturing and foregrounding participants' lived experience. Creative methods are emotionally activating and promote multiple perspectives, tolerating ambiguities and uncertainties, which are shared and even celebrated.

**Key arguments** Methods such as photovoice use imagery to elucidate narratives that are not easily captured by more traditional interview-based research techniques. However, the use of creative methods and participatory research remains novel as there is little guidance of how to navigate conceptual, practical, and analytical challenges.

**Conclusion** This paper considers these challenges, and puts forward practical and theory informed recommendations, using as study of photovoice methods for investigating ethnic inequalities in the use of the mental health act (Co-Pact) as a case study.

## PHOTOVOICE AS A RESEARCH METHOD

The use of photography and visual images as a form of research (photovoice) is an increasingly popular way to gather patient experiences that are overlooked, hidden or difficult to articulate. The purpose is to give 'voice' to experiences of injustice and adversity in a psychologically safe way. This knowledge can be used to inform the design of more inclusive policies, public services, and practice.[1]

Grounded in feminist theory and empowerment, photovoice was first developed as a method to engage communities in sharing their lived experiences in the early 1990s, and was used in areas of health education, community activism and public services.[2] In the last 30 years photovoice has been adapted for many different contexts.[1 3] Thus, the approach is gaining traction as a qualitative research method that enables nonverbal, emotional, preconscious and creative processes to be activated, in turn enabling difficult experiences to be surfaced and explored. Participation in photovoice has been demonstrated to empower participants,[4] improve reflexivity[5] and promote a sense of personal accountability.[6] Creative methods

generally, and photovoice specifically, can help to reduce power imbalances between researchers and research participants, by both parties interpreting the image and placing the participant's voice centre stage. This is especially important where there is a lack of trust, concerns about institutional harms or lack of psychological safety (as is the case for ethnic minorities in contact with mental health services), and where stigma may also prevent openness.

## IMPLEMENTING PHOTOVOICE

Photovoice methods involve posing a question and asking people to represent their views or sentiments through photographs or photo assignments, followed by a process of interpretation or photo discussion. This can be done individually or in groups, eliciting both visual (photographs) and narrative data (participants' voices). There is no convention as to the number of photo assignments or photo discussions, nor the duration of meetings and ways of assembling, analysing and presenting the resulting data. The data collection and analysis may consist of a linear approach by which participants take photographs, interpret them and share them with each other, and/or with researchers. The assumption is that material is produced and unambiguously interpreted by the participants themselves, then presented for wider discussion to generate consensus. Alternatively, some researchers have adopted a more iterative and multilayered approach whereby a series of photo assignments, and related discussions are held, with evolving meanings and reappraisal of motivations, affects, attitudes and emergent interpretations over time.[7] This permits a multitude of perspectives which may be diverse and contradictory even, but represent valuable insights into lived experience and perspectives of importance to participants.

The photographs shared as part of the photovoice process provide a rich detailed insight as to how the participants experience, interact with and understand the world around them, enabling those viewing the images to see the world from the participants' perspective. Data captured in photographs goes beyond the visual description of the content of the photograph; for example, metaphors are often used to convey feelings or experiences that are difficult to put into words. Photographs may be interpreted subjectively and are likely therefore to give expression to aspects of experience that might not be recognised, acknowledged or even shared in a group. There may be social norms to which individuals feel obliged to conform.

There are multiple levels of content, and hence, the analysis and interpretation must be appropriately nuanced. For example, the way a person chooses to compose the images, the subject matter, but also the way they engage with the task, the format (virtual or face to face), the timing and types of camera used. All of these unspoken perspectives may emerge in and impact the interpretation and analysis. Furthermore, the form of picture development, black and white, sepia or grained, and use of digital tools to modify photographs are all possible now. Undertaking the exercise in groups with other service users or peer researchers can also provide a more supportive and facilitative environment where critical reflection of the experiences and the potential for new understandings emerge from the discussion.

### Embedding rigour to maintain research integrity

As the photovoice approach gains popularity in part attributable to the significant flexibility it offers, it is important to ensure that the underpinning theory is understood and considered in both collecting data and the consequent analytic approach adopted.[8] The potential for experiential data to inform policy is progressive, and likely to be an important way of ensuring inclusive research that fully represents marginalised perspectives. However, the stages of moving through recruitment of participants, data collection, data storage, coding the data, extracting information, analysis and dissemination are less well defined. This lack of guidance is helpful if the photograph is seen as an art product, subject to multiple interpretations and meaning-making as an essential attribute that engages and promotes eudemonic satisfaction. Indeed, art in part seeks to resist reductive classifications and analysis and open the viewer and group of viewers to interact around shared and unshared experiences and interpretation.

As creative processes are not reliant on language, nor time-pressured interviews and more supportive of staged elicitation of views, experiences that are fragile and difficult to verbalise, overlooked or hidden most of the time can emerge and be deepened during the research. Visual and experience near narratives are continually available to be used in a variety of contexts, such as being curated for public display, or synthesised with evidence from other

research methods that, if used alone, risk losing insights from marginalised groups. It is important to ensure clarity of process and document decisions, so that the contributions of participants are fully represented and are not lost, diluted or redefined in the analytic process and summary statements.

The interpretation process can involve other people which can lead to the generation of additional perspectives and layers of meaning. Understanding these different layers is critically important in the analysis phase, but also to ensure authenticity in attributions of similar or different meanings.

### Analytic challenges arising from photovoice

Wang and Burris wrote 'photographs are easy to gather but difficult to analyse and summarise because they yield an abundance of complex data that can be difficult to digest' (p375).[9] Furthermore, it is not only the photographs that have been taken that are presented and discussed by participants in photo discussions, but also the photographs that they were unable to take, or chose not to take.[10] This notion of complexity is compounded by the lack of standardisation regarding how photovoice is implemented. The number of photographs being considered, the number of interactions between participants and researchers and the number of interactions between different participants can vary greatly, all of which have an impact on the volume and quality of the data collected. Yet it is unknown what the optimal volume is, and what could indicate high-quality photovoice data. These indicators may also vary according to the purpose and structure that photovoice is being conducted within. For example, shorter captions to provide context to images, may be appropriate if there are multiple iterative group discussions. Whereas longer captions may be preferable if there are fewer interactions or discussions between the participants themselves or with the researcher.

Many traditions in qualitative psychological research have been generated based on oral or written data. While the interpretation of speech or text can vary according to its intention, and the experiences of the person interpreting it, most words tend to have a clear and universally acknowledged meaning. Conversely interpreting photographs is subjective. Photographs may have multiple meanings and many different interpretations. Generally photographs have a public meaning (universally recognisable elements) and a private meaning (partial objects in the pictures that have an emotional origin and are difficult to express in words).[11 12] It is important to understand how participants assign meaning to their images and create a space that enables them to communicate their private meaning to both the researchers and other participants. This process also permits the participant, the owner of the photograph, to retain control about what is communicated and what is silenced. The photos act as an elicitation tool, to help reflection, recall and verbalisation of unexpressed views. Fostering collaboration between researchers and participants may minimise

power imbalances, and elicit new, deeper understanding as opposed to the potentially rehearsed narratives that often manifest in traditional forms of data collection such as interviews and questionnaires.

The analysis of photovoice data typically includes inductive forms of analysis such as interpretative phenomenological analysis (IPA),[13–15] grounded theory[16–18] and thematic analysis.[16 19 20] Inductive approaches use the data as a starting point for the analysis as opposed to applying a deductive framework based on existing knowledge, however 'we cannot conduct research without making assumptions' (p9).[21] There are assumptions about how knowledge is created and communicated that underpin qualitative forms of analysis. It is important to understand how these assumptions manifest and the context within which the data were collected when deciding which form of analysis is appropriate. A core component of photovoice is the critical consciousness generated by the discussion of photographs,[22] yet many of the psychological underpinnings in qualitative research are developed from a preoccupation with how an individual constructs and understands the world around them.

IPA aims to provide an in depth understanding of not only what an experience was like for participants but also how they made sense of it. Therefore, traditionally IPA involves two stages of interpretation: how the participants interpret their experiences, and how researchers interpret participants interpreting their experience.[23] The second stage risks diluting the voice of the participant. When adapting this approach for photovoice data some researchers add a data verification step[14]; others have analysed individual interviews about the photographs participants have taken using IPA, then looked for themes across these interviews[13 24] or analysed transcripts of group discussions using IPA.[15] While IPA may appear well suited for photovoice data owing to the ability view the world through the lens of the participant when examining photographs with the meaning ascribed to them by the participants themselves,[25] this approach may not be well suited to large volumes of data that contain multiple layers (such as the captions for the individual photographs and the group discussions).

Grounded theory method is an iterative form of analysis that seeks to develop new explanations concerning basic social processes.[26] This form of analysis may be useful when there are multiple photovoice workshops or photo assignment sessions, to enable the iterative development of a new theory grounded in lived experience.[27] The assumptions underpinning grounded theory also speak to participatory action research (PAR) methodology that collaborates with (rather than studies) people to develop new ways of seeing the world.[28] In practice this method may entail the development of a code book that can be applied to both visual and narrative data, alongside constant comparison and data verification with participants to ensure authenticity of the findings.[17] Interviews or focus groups may be conducted to elicit the meaning associated with the photographs, and the resulting transcripts analysed using grounded theory method.[18] This approach is particularly useful where there is capacity to hold a series of photovoice workshops or photo assignments and allow an iterative process.

Thematic analysis is described by Braun and Clarke as a method for identifying, analysing and reporting patterns within data.[29] The core aim of identifying patterns also renders this method suitable for both visual and narrative data. Thematic analysis has gained popularity over the last 15 years owing to its flexibility and accessibility for interdisciplinary researchers. The seven steps—transcription, familiarisation, coding, searching for themes, reviewing themes, defining themes and finalising themes—provide a rigorous framework that can be applied to large volumes of data in different contexts. Thematic analysis is commonly used within the psychological research approach where photovoice interviews have been employed with individuals.[19 20 30] When combining PAR and psychological research, the thematic approach is well suited to facilitating participants to collaborate in the analysis, by assisting in the identification of patterns across their data. Polytextual thematic analysis is a further development of thematic analysis principles to incorporate visual data.[31] This approach allows researchers to analyse how photographs are interpreted by participants, as opposed to being analysed and interpreted by researchers. This approach appears optimal to ensure that photographs and narratives are never analysed independently of each other.[7]

A qualitative scoping review of how photovoice was used in mental illness research found nine studies, of which seven analysed primary data.[32] The PAR approach and collaboration with the communities was central to primary studies. Grounded theory was the most cited analytical framework, however processes relating to the thematic analysis were often described in the absence of an explicitly stated analytical framework. Most studies involved participants in the analysis stage, but the majority (n=7) were still perceived to be researcher centric.[32] Within psychological research it has been proposed that the researchers and participants should analyse or interpret photographs independently, then compare their findings to generate holistic explanations and theorise phenomenon.[33] Of the nine studies only two included formal analysis of the photographs.

## CO-PACT: A CASE STUDY

Co-PACT is an National Insititute for Health and Social Care Research Policy Research Programme (NIHR PRP) funded study that seeks to combine photovoice and experience based codesign,[34] with the aim of reducing the ethnic inequalities experienced by those detained under the Mental Health Act (MHA). The use of photovoice to elucidate new perspectives and amplify the voices of marginalised populations in a creative and accessible manner renders it an optimal methodology to inform experience based codesign meetings.[35] The intention of

this study is to have an impact at a national policy level, by informing the current reform of the MHA. Photovoice data are being collected across eight NHS trusts from service users who have experience of being detained under the MHA in the last 48 months. Recruitment for these workshops has been purposive to ensure that those from marginalised backgrounds are facilitated to engage with the research process and to ensure that a range of different section types, demographics and experiences are captured.

We have linked photovoice to experience-based codesign to ensure that service users themselves (working alongside other stakeholders) will have the power or ability to change an intrinsically complex system, that has resulted in their unequal treatment. We want to ensure that we allow for their authentic and undiluted voice to be heard in spaces where those who have the power to change the system can not only hear it but also engage with it. When discussing the photographs that they provide participants critically reflect on their own situation and produce materials that can be used to plan, inform and motivate change.[36]

It is important for our analysis to capture the critical consciousness that results from the group dynamics (not solely analysis of the visual data and associated captions), to inform experience based codesign workshops which will aim to elicit changes in local systems, and consequently inform national policy.

Our approach to photovoice consists of a series of three workshops (figure 1), facilitated by researchers, delivered face to face or online according to participant preference.

We have had many interdisciplinary workshops and discussions (inclusive of people who have experienced detention under the MHA) to refine our analytic approach to the photovoice data, while being cognisant of the tensions and challenges with the complexity, volume and depth of the data we have collected. In recognition that one of the aims of the study is to foreground lived experience, it is important to acknowledge that the

analysis of the photographs begins with the participant's own interpretation of their images, developing captions and assigning meaning to their image. A second stage of analysis has also occurred with participants discussing their images in the third workshop and beginning to consider similarities and differences in their experiences, identifying patterns across the data. Based on approaches taken by other researchers working with visual data, and the principles of polytextual thematic analysis[31] we propose the following steps (table 1).

When analysing photovoice data it is also important to remember the potential therapeutic impact on the participants from taking part in these workshops, and how this impacts the information they are sharing. The creative process may have an impact on how the participants themselves relate to the experiences they are sharing, offering new insights as opposed to a rehearsed narrative. For marginalised communities, photovoice may be a more accessible way to part take in research, thus it is possible that the participants are very research naive, not used to sharing their experiences and find benefit in participation.

We propose that the following questions are considered prior to analysis in future photovoice studies:
1. What does the data consist of?
    i.   How many interactions are there between the participant and the research team?
    ii.  How many images are there per participant?
    iii. How detailed is the accompanying narrative?
2. What is the relationship between the participants and those who facilitated the workshops?
3. Is there a group narrative, individual narrative or combination of both?
4. Who is analysing the data?
5. What is the analysis informing? For what purpose is the study being conducted?

Methodological advances in this area will only be possible if we continue to innovate, document and feedback our experiences of analysing visual and narrative

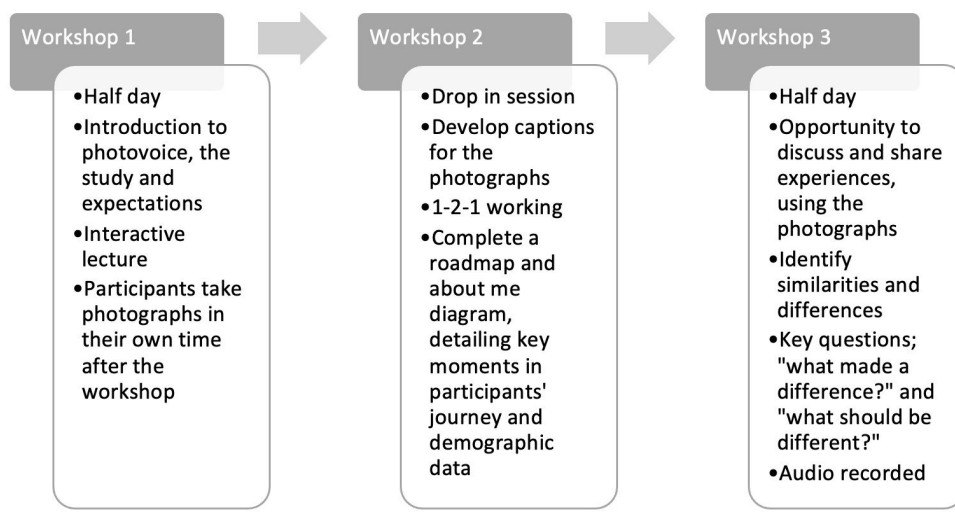

**Figure 1** The photovoice workshop process adopted in Co-PACT.

| Table 1 | Stages of analysis for multimodal data in Co-PACT |
|---|---|
| Step 1: understanding the sample | Extract demographic information from 'about me' diagrams and roadmaps into an excel sheet (Sheet 1) Initial extraction includes age, gender, ethnicity, number of times sectioned, most recent section type and diagnosis. Fields may be added as patterns in the data are recognised, eg, childhood trauma, age of onset and physical comorbidities. |
| Step 2: organise the data | Organise data into a second excel spreadsheet (Sheet 2) containing the following headings: image ID, participant's caption, image, alternative text description (this will be generated by the research team, and will be a very basic narrative description, to assist the organisation of data), mentions in transcript from the third workshop discussion, similar Images and participant demographics. This will enable a framework analysis of the data and a way to organise the data according to different variables. Researchers should keep a log book and note any observations they make during the process, such as how a group of images look together, and feelings or messages that are conveyed or noticed. |
| Step 3: individual case analysis | Consider the images and captions for each participant, consider similarities and themes within the images and the overall narrative that they are sharing. Code both the visual content of the images and the captions relating to the image, as well as noting any observations about the group of images together. |
| Step 4: exploring interactions | Go through the transcripts from the group discussions where participants discuss their images as a group, these should read as photo essays with the photographs added to the transcripts in the relevant places. Extract any references to the images and add these to Sheet 2, this should include any elements where multiple participants are discussing the same image. |
| Step 5: combining the individual and group analyses | Incorporate information from the transcripts to the existing case based analysis (Sheet 3). For a multi-site study, steps 2–5 should be done at a site level to allow an understanding of how the local context may impact the shared experiences. |
| Step 6: theme generation | Review Sheet 2 (the data) and Sheet 3 (the codes) for all sites, alongside people with lived experience, to consider and develop how codes may be grouped together to develop higher order themes. It is important that within this process, if there is a voice or experience that doesn't fit the dominant narrative that this is not disregarded or diluted in the process. We should seek to understand the exceptions, and the potential reasons for these differences as well as the main themes. |
| Step 7: theme iteration | Review higher order themes, and the image/caption combinations within those, alongside field notes from the workshops as well as group notes generated from the discussion. This is to ensure that there are not elements of the group discussion that were important to the participants who were present in the room that are missed. |
| Step 8: dissemination | Ensure that the outputs are used to motivate change, this should entail activities beyond publishing lay summaries and academic journal articles, such as presenting themes at codesign meetings and hosting public exhibitions of the images in collaboration with the participants. |

data in combination. The development of creative, participatory methodologies such as photovoice are needed to further enable marginalised populations to contribute meaningfully towards research, and inform future innovations in healthcare.

## CONCLUSION

Photovoice is a creative way of enabling participants to share difficult experiences. Initially developed for the purpose of engaging communities and motivating social justice, there are a wide range of purposes for which photovoice has now been adopted. Researchers have used photovoice to elicit the narratives of marginalised populations whose voices are often silenced. The flexibility in how photovoice is implemented renders numerous possibilities and challenges when analysing the resulting data. For researchers it is important to be transparent about the steps taken when processing the data that participants have generated and to share how decisions were informed.

**Collaborators** Co-Pact Project Team: Doreen Joseph, Nusrat Husain, Rose McCabe, Paul McCrone, Karen Newbigging, Raghu Ragavhan, Clair Dempsey, Michelle Yeung.

**Contributors** RM drafted the initial manuscript that was revised by KB, and then continual revisions made collaboratively. The content of this article reflects conversations held by the Co-PACT team, a project for which KB and RM are Co-PIs.

**Funding** This study/project is funded by the National Institute for Health Research (NIHR) Policy Research Programme (NIHR201704). The views expressed are those of the authors and not necessarily those of the NIHR or the Department of Health and Social Care.

**Competing interests** None declared.

**Ethics approval** This is a communication piece, however the Co-PACT study which is used as a case study has ethical approval from the NHS REC, and has had the protocol previously published in BMJ Open.

**Provenance and peer review** Not commissioned; externally peer reviewed.

**ORCID iD**
Roisin Mooney http://orcid.org/0000-0001-5792-3314

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
