## [Reviewer comments · BMJ Open]

ARTICLE DETAILS

TITLE (PROVISIONAL)	Analysing multi modal data that has been collected using photovoice as a research method
AUTHORS	Mooney, Roisin; Bhui, Kamaldeep; Project Team, Co-Pact

VERSION 1 – REVIEW

REVIEWER	Krishna Prasad National Institute of Mental Health and Neuro Sciences, Psychiatry
REVIEW RETURNED	26-Dec-2022

GENERAL COMMENTS	The article is well-written and provides insights into the use of photovoice in mental illness-related qualitative research, exemplified by the methods used in the Co-pact study, specifically detailing the analysis of multimodal data. I have the following suggestions and queries that the authors may consider: 1. Do they see any challenges in using this form of research in low-resource settings? Are there any recommendations they want to provide?2. Can they add any information on the use of software to analyze the qualitative data in photovoice-based research?3. Can there be potential ethical challenges in some circumstances?
---

REVIEWER	Hilary Bungay Anglia Ruskin University, Allied and Public Health
REVIEW RETURNED	02-Feb-2023

GENERAL COMMENTS	Analysing multi-modal data that has been collected using photovoice as a research method Thank you for asking me to review this Communication regarding the use of photovoice as a research method. Whilst the method has been used for a number of decades in other disciplines it is good to see creative arts-based methods adopted in the health care field, and I think for that reason it could make an important contribution to the literature. There are few points to pick up on: • I think it is worth highlighting that the method has been used widely with children and young people.• Under 'Embedding rigor to maintain research integrity' -I am not sure that the lack of guidance for the research process is true - there are authors such as Gillian Rose who have written about this. It also not that helpful for ensuring rigor in the analytic process. The first paragraph on P4, Lines 5-12 are clumsy and need re-visting. Terms such as meaning making and eudaemonic satisfaction need to be explained and/or references added. In the second and thirds
--

	paragraphs I think it would help readers if it were more specific how 'clarity of process' and 'ensure authenticity' could be achieved. Linking such statements to Table one as an example could assist this but overall, more detail regarding the rigor/credibility of the process would enhance this Communication and make it more convincing regarding the value of the method for the reader.  • Under Analytical Challenges P6 lines 31-38 'The photos act as an elicitation tool [...] as interviews and questionnaires' are there references to support these sentences? • 'Iterative process' this may need to be explained/referenced for non-qualitative researchers. • The ethical issues relating to photovoice as a method are not considered. For example (amongst others), where does copyright for the images lie if the research team want to use the images for presentations or in reports or publications? How are the identities of participants protected if they take 'selfies'- while people may originally be happy for images of themselves to be used – once on the internet they are their forever and in later years they may not want to be known as having been a service user. Likewise, if they take photos which include family, friends, members of the public. – How is this negotiated with the research participants? • Table One stages of analysis this should be renamed to include 'for the Co-pact study' –what are the 'about me' diagrams and roadmaps? I think that the different stages would be easier to follow if this was written in point form rather than narrative as it is now -
--	--

VERSION 1 – AUTHOR RESPONSE

Reviewer: 1

1. Do they see any challenges in using this form of research in low-resource settings? Are there any recommendations they want to provide?

Participatory methods and visual research is well established in high and low income countries, and can be delivered in a practical and relatively inexpensive manner. In our research we chose to use disposable cameras, and later on when adapting to an online model to facilitate engagement from different populations, participants sent in photographs via e-mail and whatsapp. Digital poverty is an issue, but digital capacity and mobiles are available in low resource settings at reasonable cost, and images and photographs or other creative works can still be produced. We consider our approach could be used in low resource settings. We have added text to this effect.

2. Can they add any information on the use of software to analyze the qualitative data in photovoice-based research?

In our work we used a combination of Excel (to organise the data) and MAXQDA (to analyse the resulting transcripts from group discussions). This has been made more explicit in the table of steps for analysis.

3. Can there be potential ethical challenges in some circumstances?

Reviewer 2 has also commented on the ethics, so we have addressed this comment below.

Reviewer: 2

- I think it is worth highlighting that the method has been used widely with children and young people.

We have added this under the photovoice as a research method section.

- Under 'Embedding rigor to maintain research integrity' -I am not sure that the lack of guidance for the research process is true - there are authors such as Gillian Rose who have written about this. It also not that helpful for ensuring rigor in the analytic process. The first paragraph on P4, Lines 5-12 are clumsy and need re-visit. Terms such as meaning making and eudaemonic satisfaction need to be explained and/or references added. In the second and thirds paragraphs I think it would help readers if it were more specific how 'clarity of process' and 'ensure authenticity' could be achieved. Linking such statements to Table one as an example could assist this but overall, more detail regarding the rigor/credibility of the process would enhance this Communication and make it more convincing regarding the value of the method for the reader.

We have changed the term guidance to flexibility, on reviewing the sentence, the message we were trying to convey is that with relation to experiential data (not necessarily only Photovoice, there are fewer established clear protocols for undertaking the work or reporting the findings, or even how to analyse the data (e.g. photovoice could be conducted using one workshop or ten). We have amended the remaining text in this section to better differentiate between the rigor required when conducting research and using creative processes as a form of public engagement.

- Under Analytical Challenges P6 lines 31-38 'The photos act as an elicitation tool [...] as interviews and questionnaires' are there references to support these sentences?

We have added the relevant references here

- 'Iterative process' this may need to be explained/referenced for non-qualitative researchers.

We have added a definition here

- The ethical issues relating to photovoice as a method are not considered. For example (amongst others), where does copyright for the images lie if the research team want to use the images for presentations or in reports or publications? How are the identities of participants protected if they take 'selfies'- while people may originally be happy for images of themselves to be used – once on the internet they are their forever and in later years they may not want to be known as having been a service user. Likewise, if they take photos which include family, friends, members of the public. – How is this negotiated with the research participants?

Much of this has been detailed in the protocol for the study that has already been published, we have included text to state this more explicitly.

- Table One stages of analysis this should be renamed to include 'for the Co-pact study' –what are the 'about me' diagrams and roadmaps? I think that the different stages would be easier to follow if this was written in point form rather than narrative as it is now –

The about me diagrams and roadmaps were how demographic information was collected in this study as opposed to using a demographic questionnaire. We are currently writing another paper on how to contextualise lived experience data, so I have removed reference to these for now to avoid confusion. Essentially this step is about extracting demographic data regardless of how it was collected.

VERSION 2 – REVIEW

REVIEWER	Krishna Prasad National Institute of Mental Health and Neuro Sciences, Psychiatry
REVIEW RETURNED	18-Mar-2023

GENERAL COMMENTS	The authors have adequately addressed my queries.
---

REVIEWER	Hilary Bungay Anglia Ruskin University, Allied and Public Health
REVIEW RETURNED	22-Mar-2023

GENERAL COMMENTS	Thank you for your careful consideration of the suggested revisions I am happy to recommend the paper for publication
---